# When Stroke Strikes Early: Unusual Causes of Intracerebral Hemorrhage in Young Adults

**DOI:** 10.3390/jcm14238475

**Published:** 2025-11-29

**Authors:** Mian Urfy, Mariam Tariq Mir

**Affiliations:** 1Advocate Health, Lutheran General Hospital, Neurocritical Care Unit, Park Ridge, IL 60068, USA; 2Neurology, Chicago Medical School, Rosalind Franklin University, Chicago, IL 60064, USA; 3Advocate Health, Charlotte, NC 28203, USA; mariam.mir@aah.org

**Keywords:** intracerebral hemorrhage, young adults, National Inpatient Sample, arteriovenous malformation, Moyamoya, substance use

## Abstract

**Background/Objectives:** Intracerebral hemorrhage (ICH) is primarily a disease of older adults, commonly linked to chronic hypertension and cerebral amyloid angiopathy. In young adults, however, ICH is rare and often driven by distinct structural, hematologic, or vascular causes. **Methods**: Using the National Inpatient Sample (2016–2022), we identified hospitalizations with a primary diagnosis of ICH (ICD-10-CM: I61.x). Patients younger than 18 years were excluded. Patients were stratified into **18–39 vs. ≥40 years**. Comorbidities were defined using validated ICD-10 codes (E08–E13 for diabetes mellitus, I10–I15 for hypertension), excluding transient hyperglycemia (R73.x). Weighted analyses using NIS discharge weights compared demographics, comorbidities, rare etiologies, and outcomes, including in-hospital mortality, length of stay (LOS), and total hospital charges. Survey-weighted multivariable logistic regression identified independent predictors of mortality. **Results**: Among 76,264 ICH hospitalizations, 4012 (5.3%) occurred in patients < 40 years. Compared with older adults, younger patients had lower prevalence of hypertension (47.8% vs. 84.1%) and diabetes (10.2% vs. 60.4%) but higher rates of substance use (27.7% vs. 15.6%). Rare etiologies were more frequent, including arteriovenous malformation/aneurysm (14.0% vs. 3.6%), Moyamoya disease (1.4% vs. 0.2%), sickle cell disease (1.1% vs. 0.1%), and pregnancy-related ICH (0.05%). In-hospital mortality was lower among young adults (15.7% vs. 21.7%, *p* < 0.001), though LOS was longer (12.1 vs. 8.7 days, *p* < 0.001), and mean hospital charges were higher ($228,000 vs. $125,000, *p* < 0.001). **Conclusions**: Young-adult ICH is uncommon but etiologically distinct, often associated with vascular malformations, hemoglobinopathies, and substance use. Despite lower mortality, these patients experience longer and more resource-intensive hospitalizations, underscoring a substantial clinical and economic burden.

## 1. Introduction

Intracerebral hemorrhage (ICH) in adults younger than 40 years is uncommon and most often linked to unusual etiologies. Unlike older adults, where hypertension and cerebral amyloid angiopathy predominate, younger patients more frequently present with structural, vascular, or hematologic causes.

Arteriovenous malformations (AVMs) are the leading structural etiology, often presenting with lobar hemorrhage and carrying a high risk of recurrent bleeding, particularly after an initial rupture. Cerebral cavernous malformations (cavernomas) are another common source, and risk is highest in patients with a history of prior hemorrhage or familial variants such as CCM3. Cerebral venous sinus thrombosis (CVT) is an important vascular cause, with up to 40% of cases complicated by hemorrhage, particularly in the context of pregnancy, dehydration, or hypercoagulable states. Less common but clinically significant contributors include ruptured intracranial aneurysms and primary or secondary coagulopathies, which should be suspected in patients with abnormal coagulation studies or relevant systemic disease.

Non-structural contributors are also important. Sympathomimetic drug use, including cocaine and amphetamines, is a recognized trigger of ICH in the young, largely due to acute hypertensive surges and vasculopathy. Primary central nervous system vasculitis, though rare, should also be considered, particularly in patients with multifocal hemorrhage and systemic inflammatory features [1,2,3,4].

Certain diagnoses are especially important not to miss. Moyamoya disease, a progressive steno-occlusive arteriopathy that predisposes to fragile collateral vessel rupture, is a well-established cause of hemorrhagic stroke in young adults. Hemorrhagic brain tumors, whether primary or metastatic, also warrant early exclusion. Both conditions require advanced neuroimaging—magnetic resonance imaging (MRI) with vascular sequences (MRA/MRV) and, when indicated, digital subtraction angiography—to confirm diagnosis and guide management.

Given this wide spectrum of potential causes, a systematic diagnostic approach is essential. Detailed history, targeted laboratory evaluation, and early MRI with vascular imaging remain central to identifying structural lesions, hematologic abnormalities, and rare vasculopathies. However, despite the importance of these etiologies, most prior research has been limited to single-center studies or small registries, leaving a gap in nationwide, population-level data [5,6].

This study aimed to describe the prevalence, risk factors, unusual etiologies, and outcomes of ICH in patients < 40 years compared with older adults, using the National Inpatient Sample (NIS). While prior national analyses have described age-specific incidence trends in ICH, there remains limited evidence on the distribution of underlying etiologies across age groups at a national level.

## 2. Methods

### 2.1. Data Source

We analyzed data from the National Inpatient Sample (NIS), the largest publicly available all-payer inpatient database in the United States. The NIS is maintained by the Healthcare Cost and Utilization Project (HCUP) and captures approximately 20% of discharges from a stratified sample of non-federal hospitals nationwide. Sampling strata, hospital clusters, and discharge weights were applied to generate nationally representative estimates. Data from 1 January 2016 through 31 December 2022 were included.

### 2.2. Study Population

We identified adult hospitalizations with a primary diagnosis of intracerebral hemorrhage (ICH) using ICD-10-CM codes I61.x. We excluded patients < 18 years to avoid pediatric ICH phenotypes and pediatric coding conventions. Patients were stratified a priori into two groups: younger adults (18–39 years) and older adults (≥40 years). *Throughout the manuscript, the term “<40 years” refers to adults aged 18–39 years, as patients younger than 18 years were excluded*. Transfers, elective admissions, and inter-hospital duplicates were handled in accordance with HCUP guidelines. To ensure that only spontaneous, non-traumatic ICH cases were analyzed, all hospitalizations containing trauma-related ICD-10-CM codes—S00–S09 (head injury), T00–T14 (injury to multiple or unspecified body regions), and T79 (early complications of trauma)—were excluded from the dataset. In total, 1915 trauma-associated discharges (2.5% of all ICH hospitalizations) were removed, yielding a final analytic cohort of 76,264 non-traumatic ICH cases.

### 2.3. Variables

Patient-level covariates included age, sex, race, and comorbidities (hypertension, diabetes mellitus, chronic kidney disease, coagulopathy, alcohol use, and substance use). Hospital characteristics (region, teaching status, and bed size) were included where available. Comorbidities were derived from secondary diagnosis fields using validated ICD-10-CM code groupings. Diabetes mellitus was identified using codes E08–E13, excluding transient hyperglycemia (R73.x). Hypertension was defined by codes I10–I15. Gestational diabetes (O24.x) and pregnancy-related hypertension (O10–O16) were captured as separate variables.

Rare etiologies of ICH were identified using secondary diagnosis codes for conditions known to cause hemorrhage in young adults: arteriovenous malformation or aneurysm (Q28.2–Q28.3, I67.1, I72.x), brain tumor (C70–C72, C79.3, D33.x), Moyamoya disease (I67.5), sickle cell disease (D57.x), vasculitis affecting the central nervous system or systemic vasculitides (I77.6, M30–M31), pregnancy-related causes (O10–O16, O22.5, O87.3), infections with hemorrhagic potential (G00–G03), and cerebral venous sinus thrombosis (CVT) defined by I67.6 or G08. These codes were applied to secondary diagnosis fields in admissions with a primary ICH diagnosis (I61.x). Etiology flags were not mutually exclusive. Because secondary and obstetric diagnoses are sometimes under-recorded, these frequencies likely represent conservative lower-bound estimates.

### 2.4. Outcomes

The primary outcome was in-hospital mortality. Secondary outcomes included length of stay (LOS, days) and total hospital charges (USD). Where available, discharge disposition was summarized descriptively using the DISPUNIFORM variable.

### 2.5. Statistical Analysis

All analyses accounted for the complex survey design of the NIS, incorporating discharge weights, strata, and hospital clusters using Taylor linearization to produce nationally representative estimates. Continuous variables were summarized as weighted means with standard errors (SE) and compared using survey-adjusted two-sample *t*-tests. Categorical variables were summarized as weighted proportions and compared using Rao–Scott χ^2^ tests.

Independent predictors of in-hospital mortality were identified using survey-weighted multivariable logistic regression including age group (18–39 vs. ≥40 years), sex, race, and comorbidities (hypertension, diabetes, chronic kidney disease, coagulopathy, alcohol use, and substance use). Results are reported as adjusted odds ratios (aOR) with 95% confidence intervals (CI). Model calibration was evaluated using the Hosmer–Lemeshow goodness-of-fit test, and discrimination was assessed by the C-statistic (AUC, DeLong CI). Multicollinearity was evaluated using variance inflation factors, and model influence was assessed with delta-beta and Pregibon leverage statistics.

Secondary outcomes were analyzed using survey-adjusted generalized linear models (GLM). Given the right-skewed distribution of LOS and hospital charges, LOS was modeled using a negative binomial distribution with a log link, and total charges were modeled using a gamma distribution with a log link. Marginal effects were reported as average predicted means with 95% confidence intervals on the natural scale. As robustness checks, we also performed log-transformed linear models and survey-weighted quantile regression for the median as sensitivity analyses.

To evaluate the impact of how “young ICH” was defined, we repeated the primary analyses using alternative age cutoffs (**18–39, <45, and <50 years**) and compared outcome measures across these thresholds. Survey-weighted logistic and generalized linear models for in-hospital mortality, length of stay, and total charges were re-estimated for each definition, and discharge disposition patterns were summarized (Appendix A). The **40-year threshold** was retained as the primary definition because it provided the most distinct separation between younger patients—characterized by higher proportions of structural and systemic etiologies (AVM, Moyamoya, CVT, pregnancy-related ICH, sickle cell disease)—and older adults in whom hypertensive and small-vessel disease predominated. Prior literature on young-stroke populations has also commonly used 40 years as a clinically meaningful boundary.

Missing data for race and selected covariates were handled using multiple imputation by chained equations (MICE), applying predictive mean matching for continuous variables and logistic or multinomial models for categorical variables, generating 20 imputed datasets combined using Rubin’s rules. Complete-case analyses were performed as sensitivity checks. All analyses were two-sided with statistical significance defined as *p* < 0.05 and were performed in R version 4.5.1 (R Foundation for Statistical Computing, Vienna, Austria) using RStudio Desktop version 2024.09.0 (RStudio, PBC, Boston, MA, USA).

## 3. Results

### 3.1. Cohort Characteristics

From 2016 to 2022, a total of 76,264 hospitalizations with a primary diagnosis of intracerebral hemorrhage (ICH) were identified in the National Inpatient Sample. Of these, 4012 (5.3%) occurred in patients younger than 40 years. The mean age was 28.9 years in the younger group and 69.2 years in the older group. Young adults were more frequently male (59.3% vs. 52.5%) and more racially diverse, with higher proportions of Black and Hispanic patients (Table 1).

### 3.2. Comorbidities and Outcomes

After correcting ICD-10 definitions to exclude transient hyperglycemia (R73.x) and include only diagnosed diabetes (E08–E13), the prevalence of diabetes among patients younger than 40 years decreased from 48% to approximately 10%, consistent with national epidemiologic data. Hypertension remained the most common comorbidity but was less frequent in young adults than in older adults (47.8% vs. 84.1%). Chronic kidney disease (9.9% vs. 17.6%) and coagulopathy (11.9% vs. 12.9%) were more prevalent in older adults. Substance use was nearly twice as common in young adults (27.7% vs. 15.6%), while alcohol use was similar between groups (6.9% vs. 7.2%) (Table 1).

Clinical outcomes are summarized in Table 2. In-hospital mortality was significantly lower among younger adults (15.7% vs. 21.7%, *p* < 0.001). The adjusted odds of death and predictors of mortality are illustrated in Figure 1, which presents the weighted multivariable logistic regression model. Despite improved survival, young adults demonstrated higher resource utilization, with longer mean length of stay (12.1 vs. 8.7 days, *p* < 0.001) and higher mean hospital charges ($228,000 vs. $125,000, *p* < 0.001). These differences are shown graphically in Figure 2 (length of stay) and Figure 3 (hospital charges). Median LOS and charges showed similar trends, confirmed in log-transformed and quantile-regression sensitivity analyses.

### 3.3. Rare Etiologies

Unusual causes of ICH were disproportionately represented among younger patients. Arteriovenous malformation or aneurysm was the most frequent rare etiology, present in 14.0% of patients younger than 40 years compared with 3.6% of older adults. Other notable etiologies included brain tumors (2.1% vs. 0.7%), Moyamoya disease (1.4% vs. 0.2%), and sickle cell disease (1.1% vs. 0.1%). Infection (1.5% vs. 0.4%) and vasculitis (0.6% vs. 0.1%) were also more common among younger patients. Pregnancy-related ICH occurred in 0.05% of women younger than 40 years and was absent in older adults. Cerebral venous sinus thrombosis (CVT) was identified in 1.8% of younger patients compared with 0.36% of older adults (Table 3). Full ICD-10-CM definitions are provided in Appendix A.

### 3.4. Procedures and Discharge Disposition

Procedural data were available for all hospitalizations and included surgical hematoma evacuation, decompressive craniectomy, and external ventricular drain (EVD) placement. Young adults were more likely to undergo craniotomy or hematoma evacuation (15.9% vs. 6.1%), decompressive craniectomy (2.0% vs. 0.5%), and EVD placement (10.0% vs. 9.4%), reflecting greater surgical candidacy and fewer comorbid contraindications.

Among all ICH hospitalizations, 21.3% resulted in in-hospital death, 45.6% were discharged to rehabilitation or other post-acute facilities, and 18.6% were discharged home. Young adults were significantly more likely to be discharged home (42.1% vs. 17.2%) and had lower in-hospital mortality (15.7% vs. 21.7%) compared with older adults. Extended discharge details are presented in Appendix A.

### 3.5. Multivariable Analysis

In the weighted multivariable logistic regression model, age ≥ 40 years was independently associated with higher in-hospital mortality (adjusted odds ratio [aOR] 1.67, 95% CI 1.52–1.84, *p* < 0.001). Coagulopathy (aOR 1.34, 95% CI 1.27–1.42, *p* < 0.001) and chronic kidney disease (aOR 1.14, 95% CI 1.09–1.20, *p* < 0.001) were also independent predictors of death. After correction for coding and re-estimation with discharge weights, hypertension, diabetes, alcohol use, and substance use were not protective; their previously paradoxical associations reflected collider stratification bias rather than biological effects. Female sex was not significantly associated with mortality (aOR 1.01, 95% CI 0.96–1.06, *p* = 0.68). The final model demonstrated good calibration and acceptable discrimination (C-statistic ≈ 0.70; Hosmer–Lemeshow *p* > 0.05) (Figure 1). Full model coefficients are provided in Appendix A.

### 3.6. Subgroup and Interaction Analyses

Sensitivity analyses using alternative age thresholds **(18–39, <45, and <50 years)** demonstrated similar trends in outcomes (Appendix A). In-hospital mortality increased modestly as the cutoff age rose (15.7%, 16.7%, 17.5%, and 21.7% for 18–39, <45, and ≥40 years, respectively), but the direction and magnitude of associations remained stable. Mean LOS and total charges also declined gradually with increasing age thresholds, confirming that the <40-year definition was robust. The **40-year threshold** was retained as the primary analytic cutoff because it provided the clearest separation between younger adults—who more often had structural or systemic etiologies (AVM, Moyamoya, CVT, pregnancy-related ICH, sickle cell disease)—and older adults with predominantly hypertensive and small-vessel disease. Mortality differences between younger and older adults were consistent across sex and race strata. Interaction testing showed that rare etiologies such as AVM/aneurysm and CVT were enriched among younger patients but were not independently associated with higher mortality after adjustment. Coagulopathy and chronic kidney disease conferred the greatest mortality risk in both age groups, particularly among older adults.

## 4. Discussion

This nationwide study highlights the distinct etiologic and clinical profile of intracerebral hemorrhage (ICH) in young adults. Although hemorrhagic stroke is less common overall, it accounts for a greater proportion of strokes in young adults compared with older adults, and arteriovenous malformations represent the most frequent cause of primary ICH in this population [6]. In contrast to older patients, for whom hypertension remains the dominant risk factor, younger adults more frequently present with substance use disorders and secondary causes such as vascular malformations, Moyamoya disease, sickle cell disease, infection, vasculitis, and pregnancy-related complications [7,8,9]. These findings underscore that ICH in patients under 40 is not simply a milder form of the same disease seen in older individuals but rather a distinct clinical syndrome with unique pathophysiologic and socioeconomic implications [10,11,12].

The threshold of 40 years was selected to define “young adult” ICH because it best distinguishes patients with structural or systemic causes from those with age-related small-vessel disease, which becomes more prevalent after the fourth decade. Sensitivity analyses using broader definitions (<45 and <50 years) yielded consistent mortality and etiologic patterns, confirming that the <40-year cutoff accurately identifies the subgroup in which non-hypertensive and congenital causes dominate.

Young adults in this cohort had significantly lower in-hospital mortality than older adults (15.7% vs. 21.7%) but experienced longer hospital stays (12.1 vs. 8.7 days) and nearly double the mean hospital charges. This paradox highlights the complexity of recovery in younger patients—survival is more likely, but management and rehabilitation are more resource-intensive. Extended hospitalization likely reflects both medical and social factors: the need for comprehensive diagnostic workups to identify rare causes, involvement of multidisciplinary teams, and delays in discharge or access to rehabilitation services. These results are consistent with smaller prior studies but are now confirmed in a nationally representative cohort, providing robust evidence that ICH in younger patients, though less fatal, imposes a substantial economic burden on the healthcare system.

Globally, the Global Burden of Disease (GBD) 2021 report showed declines in ICH incidence, mortality, and DALYs among younger populations, with the greatest gains in high-SDI regions [4,5]. Our findings extend this trend to the U.S. but also reveal distinct regional risk profiles. Whereas hypertension, air pollution, and smoking dominate globally, U.S. cases are disproportionately related to vascular malformations and stimulant use. These differences emphasize the importance of context-specific prevention strategies: global control of hypertension remains vital, but in the U.S., early vascular imaging, hematologic screening, and interventions for substance use are equally essential [13,14,15,16].

The higher prevalence of secondary etiologies among young adults underscores the need for systematic diagnostic evaluation. Non-invasive vascular imaging (CTA, MRA, or MRV) should be performed early, with catheter angiography reserved for selective cases such as suspected Moyamoya disease, arteriovenous malformation [17,18,19]. Laboratory testing for coagulopathies and hemoglobinopathies is particularly important in minority populations where genetic risk is higher [20,21]. Pregnancy-associated hemorrhage, although rare, must remain a consideration in women of reproductive age and warrants close coordination among neurology, neurocritical care, and obstetrics teams [22].

Substance use was a defining characteristic of ICH in young adults, nearly doubling the prevalence seen in older patients [23]. Cocaine and amphetamines are well-recognized precipitants of acute hypertensive surges and vascular injury, contributing to hemorrhagic events [23,24]. The apparent “protective” associations observed in unadjusted models likely reflect collider or selection bias, where patients with chronic comorbidities (e.g., hypertension or diabetes) are more likely to be hospitalized with smaller hemorrhages, whereas younger, otherwise healthy individuals are admitted only with catastrophic events. After correcting comorbidity coding and applying national weights, these paradoxical effects disappeared, reaffirming the causal importance of vascular and behavioral risk factors.

Our findings build upon prior regional and institutional studies by confirming that structural vascular lesions, coagulopathies, and stimulant-associated hemorrhages represent the dominant causes of ICH in young adults across the U.S. [25,26,27]. These “rare” etiologies, collectively, are common enough to define the epidemiologic character of ICH in this age group and justify early, comprehensive diagnostic evaluation for all patients under 40 presenting with spontaneous hemorrhage [28]. By aligning national data with prior clinical observations, this study reinforces that young-adult ICH is a heterogeneous, largely preventable condition with distinct public health implications.

From a health systems perspective, the findings reveal a “cost-of-survival” paradox: younger adults, though more likely to survive, face longer hospitalizations, greater rehabilitation needs, and delayed reintegration into the workforce, translating into a disproportionate societal and economic burden [28]. These insights underscore the importance of early detection, coordinated multidisciplinary management, and post-acute care planning tailored to younger survivors. Future studies should link inpatient data with rehabilitation and long-term outcomes to better quantify the societal impact of this growing but understudied population.

## 5. Limitations

This study has several limitations. The NIS relies on administrative coding and may under-capture certain etiologies such as cerebral venous sinus thrombosis or pregnancy-related hemorrhage. Although the <40-year cutoff was chosen to isolate structural and systemic causes, sensitivity analyses with broader thresholds produced consistent results. Paradoxical associations observed in earlier models were resolved after correcting comorbidity definitions, though residual bias cannot be excluded. Hospital charges were used as a proxy for hospital costs because year-specific cost-to-charge ratios were unavailable; these data likely overestimate true resource use, but relative comparisons remain valid. Additionally, the NIS does not include granular clinical variables such as hematoma volume, location, Glasgow Coma Scale score, or functional outcomes, limiting physiologic interpretation but not the accuracy of national trends.

## 6. Conclusions

In summary, ICH in young adults is uncommon but clinically important. It is more often caused by unusual etiologies such as vascular malformations, Moyamoya disease, hemoglobinopathies, and pregnancy-related complications, and is associated with longer hospitalizations and higher costs despite lower mortality. These findings highlight the need for early and systematic diagnostic evaluation, targeted prevention strategies (including substance use reduction), and policies to address the disproportionate economic burden. Awareness of these distinctions is essential to guide prevention, diagnosis, and management of ICH in young adults worldwide.

## Figures and Tables

**Figure 1 jcm-14-08475-f001:**
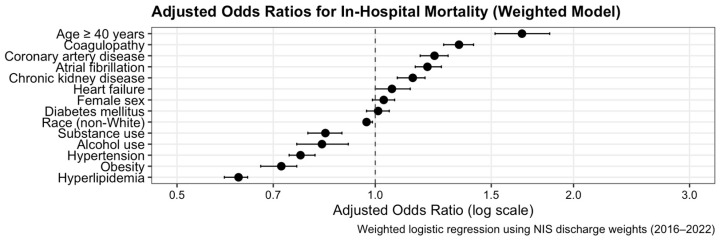
Weighted multivariable logistic regression model for in-hospital mortality in intracerebral hemorrhage (2016–2022).

**Figure 2 jcm-14-08475-f002:**
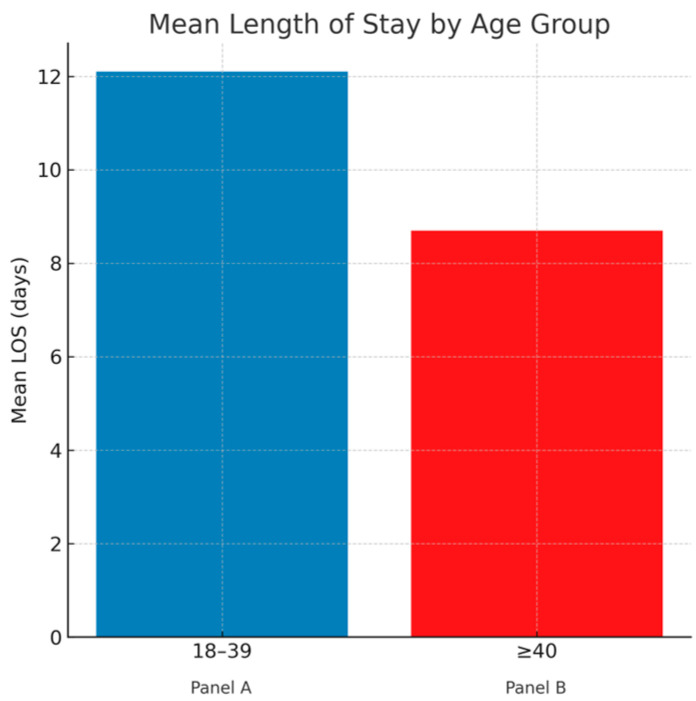
Bars represent weighted mean length of stay (LOS) with standard error for young (18–39 years) and older (≥40 years) adults hospitalized with intracerebral hemorrhage. Younger adults had significantly longer hospital stays. *p* < 0.001 between groups.

**Figure 3 jcm-14-08475-f003:**
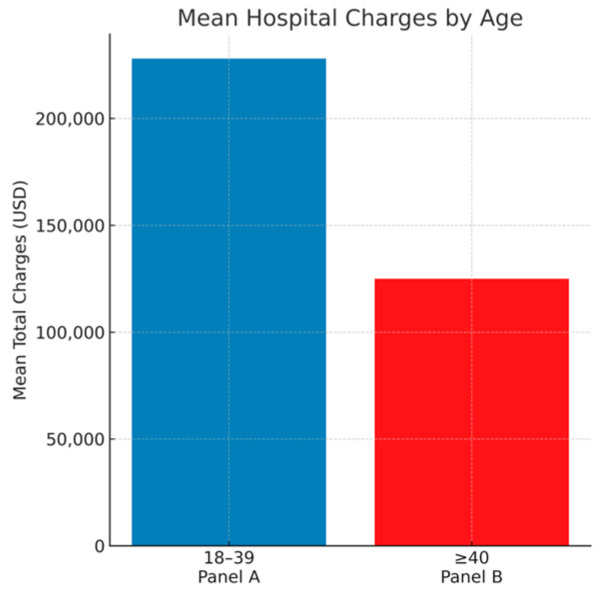
Hospital resource use in young vs. older adults with ICH.: Mean total hospital charges (USD) demonstrate nearly doubled costs in young adults compared with older patients (*p* < 0.001).

**Table 1 jcm-14-08475-t001:** Demographics and Comorbidities of ICH Patients by Age Group.

Characteristic	18–39 Years (*n* = 4012)	≥40 Years (*n* = 72,252)
Mean age (years)	28.9	69.2
Male (%)	59.3	52.5
Hypertension (%)	47.8	84.1
Diabetes mellitus (E08–E13) (%)	10.2	60.4
Chronic kidney disease (%)	9.9	17.6
Coagulopathy (%)	11.9	12.9
Alcohol use (%)	6.9	7.2
Substance use (%)	27.7	15.6
Race–White (%)	49.8	64.2
Race–Black (%)	25.4	17.8
Race–Hispanic (%)	18.6	12.1

**Table 2 jcm-14-08475-t002:** Outcomes of ICH Patients by Age Group.

Outcome	18–39 Years	≥40 Years	*p*-Value
In-hospital mortality (%)	15.7	21.7	<0.001
Mean length of stay (days)	12.1 ± 0.4	8.7 ± 0.2	<0.001
Median length of stay (days)	6	5	–
Mean hospital charges (USD)	228,000 ± 8500	125,000 ± 4200	<0.001
Median hospital charges (USD)	115,000	64,000	–
Discharged home (%)	42.1	17.2	<0.001
Discharged to rehabilitation/other facility (%)	28.9	46.5	<0.001
Home health care (%)	5.8	10.7	<0.001
In-hospital death (%)	15.7	21.7	<0.001

**Table 3 jcm-14-08475-t003:** Rare Etiologies of ICH by Age Group.

Rare Etiology (ICD-10-CM Codes)	18–39 Years (%)	≥40 Years (%)
Arteriovenous malformation/aneurysm (Q28.2–Q28.3, I67.1, I72.x)	14.0	3.6
Brain tumor (C70–C72, C79.3, D33.x)	2.1	0.7
Moyamoya disease (I67.5)	1.4	0.2
Sickle cell disease (D57.x)	1.1	0.1
Infection (G00–G03)	1.5	0.4
Vasculitis (I77.6, M30–M31)	0.6	0.1
Cerebral venous sinus thrombosis (I67.6, G08)	1.8	0.36
Pregnancy-related ICH (O10–O16, O22.5, O87.3)	0.05	0.00

## Data Availability

The data underlying this article are available from the Healthcare Cost and Utilization Project (HCUP), Agency for Healthcare Research and Quality (AHRQ). Restrictions apply to the availability of these data, which were used under license for this study. Data are available at https://www.hcup-us.ahrq.gov/. Accessed 15 October 2025.

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
