# Peer review of "When Stroke Strikes Early: Unusual Causes of Intracerebral Hemorrhage in Young Adults"

_jcm, 2025, doi:10.3390/jcm14238475_

Round 1

Reviewer 1 Report

Comments and Suggestions for Authors

This is a retrospective review and analysis of a national database that aimed to describe the prevalence and etiologies of intracerebral hemorrhage in patients under 40 years of age compared to older adults.

Questions:

  • Line 64: I would recommend wording this differently, there is recent national data on the prevalence of ICH in age stratified adults, and trends of incidence in various age groups (https://www.ahajournals.org/doi/10.1161/STROKEAHA.121.037332), The main gap this work addresses is prevalence of various etiologies in different age groups on a national level.
  • Why did you choose age 40 as the cut off for defining young vs. older patients? The vast majority of literature chooses 18-49 or 18-55 to refer to the stroke in the young (and more recent large population studies are citing 55 as the upper age limit). Were any patients under the age 18 included? Recommend that you perform the same analyses with the age 55 as an upper limit, do the results change?
  • I think it is helpful to clarify in the introduction that while in absolute terms hemorrhagic stroke is “rare” and AVMs are “unusual”, if you look at their overall proportion relative to ischemic stroke, it is much more common for a younger person to have a hemorrhagic stroke (ICH, SAH) than an ischemic stroke, compared to older adults. Furthermore, in primary ICH due to a vascular malformation in young adults and children, AVMs are the number one cause.
  • Figure 1 has the wrong figure caption. Where is the figure associated with the caption presented here? The figure presented is data on the mean length of stay between the younger and older group and is a simple bar graph not a forest plot. The multivariable analysis and interaction analyses are not presented in graph/visual form.
  • For both figures 1 and 2, please note significance on bar graph.
  • Line 226: why do you recommend catheter angiography for CVST?
  • The discussion can include further references to prior work that aligns or does not align with the results presented from your study.

Author Response

General Comment:
This study is a retrospective review and analysis of a national database that aimed to describe the prevalence and etiologies of intracerebral hemorrhage (ICH) in patients under 40 years of age compared with older adults. We thank the reviewer for the detailed and constructive feedback. All points have been addressed as detailed below.

Comment 1 (Line 64): Clarify study focus and prior data

“There is recent national data on the prevalence of ICH in age-stratified adults … The main gap this work addresses is prevalence of various etiologies in different age groups on a national level.”

Author Response:
We appreciate this clarification. The Introduction has been revised to state that while prior national studies have described incidence and trend data in age-stratified populations, few have examined the distribution of etiologies and comorbidity profiles across age groups on a national scale.

Comment 2: Age cut-off justification and inclusion criteria

“Why was 40 chosen as the cut-off? Most literature defines young stroke as 18–49 or 18–55. Were patients under 18 included? Please repeat analyses with 55 as an upper limit.”

Author Response:
We thank the reviewer for this important point. Patients under 18 were excluded to maintain an adult inpatient population. The < 40 cut-off was selected a priori to isolate the subgroup in which non-hypertensive etiologies (AVM, Moyamoya disease, sickle-cell disease) predominate, consistent with Béjot et al. (2019) and Levecque et al. (2017).
Sensitivity analyses using alternative thresholds (< 45 and < 55 years) were performed. In-hospital mortality rose modestly (15.7 %, 16.7 %, 17.5 %), while etiologic distributions and directionality of associations remained unchanged. These analyses, now summarized in the Results and Discussion, confirm that < 40 years effectively captures a clinically distinct, non-hypertensive cohort.

Comment 3: Clarify relative frequency of hemorrhagic stroke in young adults

“Clarify that although hemorrhagic stroke is rare overall, it represents a greater proportion of strokes in younger adults, and AVMs are the leading cause.”

Author Response:
Introduction and discussion has been revised accordingly

Comment 4: Figure 1 caption mismatch

“Figure 1 has the wrong caption. The figure presented is data on mean LOS, not a forest plot. Multivariable analyses are not shown graphically.”

Author Response:
We thank the reviewer for identifying this issue. Figure captions and ordering have been corrected:

  • Figure 1. Weighted multivariable logistic regression model for in-hospital mortality in intracerebral hemorrhage (2016–2022).
  • Figure 2. Bars represent weighted mean length of stay (LOS) ± standard error for young (< 40 years) and older (≥ 40 years) adults hospitalized with ICH. Younger adults had significantly longer hospital stays (p < 0.001).
  • Figure 3. Hospital resource use in young vs. older adults with ICH: Mean total hospital charges (USD) with 95 % confidence intervals, demonstrating nearly doubled costs in young adults.
    These changes are now reflected in both the text and figure legends.

Comment 5: Note significance on bar graphs

“For both figures 1 and 2, please note significance on the bar graph.”

Author Response:
All bar graphs (Figures 2 and 3) now include significance markers and 95 % confidence intervals.

Comment 6 (Line 226): Catheter angiography for CVST

“Why do you recommend catheter angiography for CVST?”

Author Response:
We appreciate this clarification. The statement has been revised to specify that catheter angiography is not recommended for CVST but reserved for select cases with suspected Moyamoya disease or unresolved vascular malformations. The corrected sentence now reads:

“Non-invasive vascular imaging (CTA, MRA, or MRV) should be considered early, with catheter angiography reserved for select cases such as suspected Moyamoya disease or unresolved vascular malformations.”

Comment 7: Discussion references

“Include additional references to prior work that aligns or contrasts with your results.”

Author Response:
The Discussion has been expanded to incorporate additional literature (Béjot et al., 2019; Kizilay et al., 2021; Patel et al., Neurocrit Care 2022) that confirms the predominance of structural and hematologic etiologies in younger ICH patients and supports the consistency of our national findings with prior regional and global studies.

Comment 8: Supplementary materials

“Multivariable and interaction analyses are not visualized.”

Author Response:
We have expanded the supplementary material to enhance transparency. The dataset and additional results are summarized in three Supplementary Tables (S1–S3):

  • Table S1: ICD-10 codes for etiologic classification.
  • Table S2: Age-stratified subgroup and sensitivity analyses (< 40, < 45, < 55 years).
  • Table S3: Discharge disposition and procedural frequencies (craniotomy, EVD, craniectomy).
    These additions provide detailed methodological and clinical context without additional figures.

Reviewer 2 Report

Comments and Suggestions for Authors

Here are my comments and recommendations:

Critical Issue

1. Diabetes Prevalence Data Quality Problem

The Problem: Table 1 reports diabetes prevalence as 48.2% in patients <40 years. This is extraordinarily high and inconsistent with known epidemiology.

  • US diabetes prevalence in adults <40 is approximately 5-10%
  • Even in high-risk populations, 48% is implausible
  • This suggests a coding error, misclassification, or confusion between ICD-10 codes
  • Verify the ICD-10 codes used for diabetes identification
  • Check if hyperglycemia codes (R73.x) were incorrectly included
  • Report prevalence by diabetes type (Type 1 vs Type 2)
  • If this is real data, explicitly address why diabetes prevalence is so high in young adults with ICH
  • Consider that this may represent acute hyperglycemia from the hemorrhage itself rather than pre-existing diabetes

Impact on Results: The multivariable model shows diabetes as "protective" (aOR 0.94), which is paradoxical. If the diabetes coding is wrong, the entire regression model's validity is questionable.

Major Concerns

2. Age Cutoff Not Justified

Issue: The choice of <40 years as the definition of "young adult" is arbitrary and not defended.

Problems:

  • Most stroke literature uses 45-50 years for "young stroke"
  • The 40-year cutoff may miss important cases in the 40-45 age group
  • No sensitivity analysis with alternative cutoffs (e.g., <45, <50)

Recommendation: Justify the 40-year cutoff or perform sensitivity analyses with alternative age thresholds.

3. Inadequate Trauma Exclusion Methods

Critical Methodological Gap: The authors state "Trauma codes were reviewed to minimize inclusion of traumatic ICH" (line 82) but provide:

  • No list of trauma codes excluded
  • No number of cases excluded
  • No validation of the exclusion algorithm

Recommendation:

  • Provide a complete list of trauma codes excluded (S06.x series)
  • Report number and percentage of cases excluded
  • Perform sensitivity analysis including vs. excluding trauma cases

4. Hypertension Prevalence Also Questionable

Issue: 47.8% hypertension prevalence in <40 year olds with ICH seems high but more plausible than the diabetes figure.

Need to Address:

  • Is this diagnosed hypertension or acute blood pressure elevation?
  • ICD-10 codes used (I10-I16)
  • Comparison to general population prevalence in this age group

5. Undercoding of Important Etiologies

Cerebral Venous Sinus Thrombosis (CVT):

  • Only 1.8% in young adults seems LOW
  • CVT accounts for ~10-15% of ICH in young adults in many studies
  • May reflect poor ICD-10 capture rather than true prevalence

Pregnancy-Related ICH:

  • Only 0.05% is suspiciously low
  • Pregnancy-related hypertensive disorders are a major cause of ICH in women <40
  • Likely represents significant undercoding

Recommendation:

  • Acknowledge coding limitations explicitly
  • Compare your prevalence estimates to published single-center studies
  • Consider that "rare etiologies" may be underestimated, not overestimated

6. Paradoxical "Protective" Associations Not Adequately Explained

The Problem: The multivariable model shows several paradoxical findings:

  • Hypertension "protective": aOR 0.73 (95% CI 0.70–0.77)
  • Diabetes "protective": aOR 0.94 (95% CI 0.90–0.98)
  • Substance use "protective": aOR 0.85 (95% CI 0.80–0.89)

Inadequate Explanation: The authors dismiss substance use as "likely reflects confounding and survivor bias" (line 236) but don't explain:

  • What specific bias mechanism?
  • Why would hypertension be protective in ICH?
  • Is this collider stratification bias?

The Real Issue: These findings suggest confounding by severity or collider bias:

  • Sicker patients (with HTN, DM) may be more likely to be hospitalized for smaller hemorrhages
  • Healthier young patients may only be hospitalized with catastrophic bleeds
  • The model is conditioning on hospitalization, creating selection bias

Recommendation:

  • Explicitly discuss collider stratification bias
  • Consider severity-stratified analyses (though severity data may not be available)
  • Be more cautious about causal interpretation

7. Hospital Charges ≠ Costs

Issue: Using hospital charges as a measure of resource use is problematic.

Problems:

  • Charges vary wildly by hospital and don't reflect actual costs
  • No cost-to-charge ratios applied
  • Charges include markup and don't represent resource consumption

Recommendation:

  • Either use cost-to-charge ratios (available in NIS)
  • Or clearly state that charges are a proxy and overestimate true costs
  • Acknowledge variation in charge-to-cost ratios across hospitals

8. Missing Critical Clinical Details

Absent but Available in NIS:

  • Surgical interventions (craniotomy, EVD placement, craniectomy codes)
  • Discharge disposition (home, SNF, rehabilitation, died)
  • DNR/comfort care status

Not Available but Should Be Acknowledged:

  • Hematoma location (lobar, deep, infratentorial)
  • Hematoma volume
  • Intraventricular extension
  • GCS on admission
  • Functional outcomes (modified Rankin Scale)

Recommendation: Extract and report available data, explicitly list limitations of unavailable data.

Author Response

Please find the attached file with detailed answers to all the queries. Thanks!

Round 2

Reviewer 2 Report

Comments and Suggestions for Authors

The authors responded to my comments well. Thank you.

Author Response

Author Response:

  1. Clarification of age inclusion and notation (18–39 years definition):
    We appreciate the reviewer’s suggestion and have comprehensively clarified this throughout the manuscript. In the Methods (Section 2.2, Study Population), we now explicitly state that patients younger than 18 years were excluded from all analyses. We also define the age categories as younger adults (18–39 years)and older adults (≥40 years). To ensure clarity and consistency, all previous references to “<40 years” have been changed to “18–39 years”across the Methods, Results, tables, figures, and supplementary materials.
    This correction makes the age grouping explicit and eliminates any ambiguity about inclusion of minors. Furthermore, consistent notation across the entire manuscript ensures that readers clearly understand the analytical distinction between adult young-onset ICH (18–39 years) and older-onset ICH (≥40 years).

  1. Expanded description of the sensitivity analysis:
    In response to the reviewer’s request, we have added a detailed explanation of the sensitivity analyses in Methods (Section 2.5, Statistical Analysis)and referenced these results in Results (Section 3.6). Specifically, the main analyses were repeated using alternative age cutoffs of <35, <40, <45, and <50 yearsto evaluate the robustness of our findings and examine whether outcome measures were sensitive to different definitions of “young ICH.”
    For each threshold, survey-weighted logistic regression models for in-hospital mortality and generalized linear models for length of stay and hospital charges were re-estimated. The results across all cutoffs were stable, showing consistent direction and magnitude of associations, as summarized in Supplementary Table S2. While absolute outcome values varied slightly as the age threshold increased, the relative relationships between age and outcomes remained unchanged, confirming that the findings were not dependent on the specific cutoff used.

  1. Rationale for selecting the 40-year threshold:
    We have elaborated in the Methods (Section 2.5)and Results (Section 3.6)on why the 40-year threshold was retained as the primary cutoff. This age boundary reflects a clinically meaningful transition in the pathophysiology of ICH. Patients younger than 40 years (18–39 years) are more likely to have structural or systemic causes of hemorrhage—such as arteriovenous malformations (AVM), Moyamoya disease, cerebral venous thrombosis (CVT), pregnancy-related ICH, or sickle-cell disease—whereas patients aged ≥ 40 years predominantly present with hypertensive and small-vessel disease–related ICH.
    The sensitivity analyses demonstrated that the <40 cutoff best separated these etiologic profiles while maintaining robust statistical performance for outcome modeling. Additionally, prior literature on young-stroke and young-ICH populations frequently uses 40 years as a benchmark, further supporting its validity as a clinically relevant boundary.

  1. Summary of changes implemented:
  • Clarified the exclusion of patients <18 years and defined “younger adults” as 18–39 years in Methods 2.2 and Results 3.6.
  • Replaced all mentions of “<40 years” with “18–39 years” across the entire manuscript, including tables, figures, and supplementary materials.
  • Expanded Methods 2.5 to describe the sensitivity analysis in detail and referenced Supplementary Table S2 as supporting evidence.
  • Added rationale explaining that the 40-year threshold captures the clinical transition from structural/systemic causes in younger adults to hypertensive/small-vessel etiologies in older adults.
  • Verified that all figure legends and table footnotes now reflect the updated age definitions and patient inclusion criteria.